# Neglecting Plant-microbe Symbioses Leads to Underestimation of Modeled Climate Impacts

Mingjie Shi [1,2], Joshua B. Fisher [1,2], Richard P. Phillips [3], Edward R. Brzostek [4*]

Jet Propulsion Laboratory, California Institute of Technology, 4800 Oak Grove Drive, Pasadena, CA 91109, USA
Joint Institute for Regional Earth System Science and Engineering, University of California at Los Angeles, Los Angeles, CA 90095, USA
Department of Biology, Indiana University, 702 N. Walnut Grove Avenue, Bloomington, IN 47405, USA
Department of Biology, West Virginia University, 53 Campus Drive, Morgantown, WV 26506, USA

Submitted to: *Biogeosciences*

* Corresponding author email address: erbrzostek@mail.wvu.edu

**Keywords**: Mycorrhizal Fungi; Nitrogen limitation Climate Change; Community Atmosphere Model; Community Land Model, Fixation and Uptake of Nitrogen

**Abstract**

The extent to which terrestrial ecosystems slow climate change by sequestering carbon hinges in part on nutrient limitation. We used a coupled carbon–climate model that accounts for the carbon cost to plants of supporting nitrogen-acquiring microbial symbionts to explore how nitrogen limitation affects global climate. To do this, we first calculated the reduction in net primary production due to the C cost of N acquisition. We then used a climate model to estimate the impacts of the resulting increase in atmospheric $CO_2$ on temperature and precipitation regimes. The carbon costs of supporting symbiotic nitrogen uptake reduced net primary production by 8.1 Pg C $yr^{-1}$, with the largest absolute effects occurring in tropical forest biomes and the largest relative changes occurring in boreal and alpine biomes. Globally, our model predicted relatively small changes in climate due to the C cost of N acquisition with temperature increasing by 0.1°C and precipitation decreasing by 6mm $yr^{-1}$. However, there were strong regional impacts with the largest impact occurring in boreal and alpine ecosystems, where such costs were estimated to increase temperature by 1.0 °C and precipitation by 9 mm $yr^{-1}$.; as such, our results suggest that carbon expenditures to support nitrogen-acquiring microbial symbionts have critical consequences for Earth's climate, and that carbon–climate models that omit these processes will over-predict the land carbon sink and under-predict climate change.

## 1. Introduction

The magnitude of carbon (C) uptake by the terrestrial biosphere strongly depends on the availability of nutrients to support net primary production (NPP) (Zaehle *et al.* 2015; Wieder *et al.* 2015; Wang *et al.* 2015). Most soil nutrients exist in unavailable forms and consequently plants must expend a portion of their assimilated C on nutrient acquisition (Johnson, 2010; Mohan *et al*. 2014). Many plants allocate up to 20% of their C to support symbiotic mycorrhizal fungi, which can be responsible for almost half of plant nitrogen (N) uptake in ecosystems (Hobbie, 2006; Högberg & Högberg, 2002; Parniske, 2008) or to support symbiotic N-fixing bacteria (Shi *et al.* 2016). Given the magnitude of these C expenditures, Earth System Models (ESMs) that do not account for the costs of supporting symbiotic microbes may overestimate NPP and the ability of terrestrial ecosystems to slow climate change.

Nearly all land plants have evolved symbiotic strategies for coping with nutrient limitation. Plant associations with mycorrhizal fungi such as arbuscular mycorrhizae (AM) and ectomycorrhizae (ECM), or with N-fixers, are critical for the uptake of soil nutrients and as such, impact C and nutrient cycling (Phillips *et al*. 2013; Wurzburger *et al*. 2017). Recent data syntheses have shown that ECM and AM ecosystems have divergent C-nutrient economies that respond differently to elevated $CO_2$ and N deposition (Canham & Murphy, 2017; Terrer *et al*. 2016; Terrer *et al*. 2017). Despite this, the C cost for nutrient acquisition remains largely absent in most C–climate models which assume that plants do not expend C to take up N and that NPP is only downregulated if there is not enough N to support biomass synthesis. As such, there have been few first order estimates of the extent to which variable plant investment in strategies that facilitate N uptake can impact rates of climate change. Shi *et al*. (2016) showed that dynamically predicting and accounting for the C cost of N acquisition reduced global NPP by 13%, and thus, models that assume N uptake requires no C expenditures potentially underestimate the rate of atmospheric $CO_2$ rise. These results not only underscore the importance of including the C cost of symbiotic microbes in ESMs but also highlight the critical role that plant-microbe interactions play in mediating rates of climate change.

Global C-climate models represent the scientific community's integrated hypotheses on how climate responds to anthropogenic forcing. In addition to forecasting climate, ESMs can be used to perform "experiments" at spatial and temporal scales that are logistically unfeasible to identify important feedbacks and processes in the Earth's climate system (Fisher *et al*. 2014). Accordingly, our objective was to explore the potential feedbacks between the C cost of supporting symbiotic N acquisition with climate by performing model experiments with and without these costs in a C-climate model. To streamline the complexity of the Earth-scale computations, we used the Community Atmosphere Model (CAM) with prescribed sea surface temperatures and sea ice and a version of the Community Land Model (CLM) which predicts the impacts of symbiotic processes on coupled C and N dynamics. We are focusing on the dynamic processes between the land and atmosphere, and this C-climate model assessment represents the first effort to determine the sensitivity of the Earth's climate system to plant-microbe symbiotic interactions.

## *2*. Material and Methods
### 2.1 Models

We used the Fixation and Uptake of Nitrogen (FUN) sub-model to dynamically compute the C cost and N benefit of AM fungi, ECM fungi, and N-fixers. FUN optimally allocates the C gained from NPP to N acquisition through the following pathways: uptake from soil (via AM or ECM roots, or non-mycorrhizal roots), retranslocation from senescing leaves, and symbiotic biological N fixation (Brzostek *et al*. 2014; Fisher *et al*. 2010). FUN then down-regulates NPP based upon the integrated C cost across each pathway and how much N was acquired to fix C into biomass. The C cost of each pathway is calculated using functions that relate costs to drivers with soil uptake a function of soil N concentration and root biomass, retranslocation a function of leaf N, and fixation a function of temperature (Brzostek *et al*. 2014; Shi *et al*. 2016). In FUN, AM plants benefit when N is relatively abundant, ECM plants benefit when N is strongly limiting, and N-fixers thrive in high energy environments with high N demand (Brzostek *et al*. 2014).

We used the Community Land Model version 4 (CLM) (Lawrence *et al*. 2011; Oleson *et al*. 2010). CLM is a terrestrial biosphere model that predicts the impacts of greenhouse gases and meterological conditions on the land surface's energy, carbon, and water budgets. Importantly, CLM includes coupled C and N cycles whereby the internal recycling, loss, and inputs of N in the soil pool are dynamically modeled to predict the avaialbility of N to support plant biomass synthesis (Lawrence *et al*. 2011; Oleson *et al*. 2010).

FUN was recently coupled into CLM (CLM-FUN) with model simulations showing that the C cost of N acquisition reduces the C sink strength of the terrestrial biosphere (Shi *et al*. 2016). CLM-FUN predicts the C cost of N acquisition from the soil by ectomycorrhizal, arbuscular mycorrhizal, and nonmycorrhizal roots based upon root biomass (a proxy for access) and soil nitrogen concentrations (a measure of availability of N for plants to take up). Previously, the parameter controlling the sensitivity of the C cost of N acquisition to root biomass was low. As such the C cost of N acquisition showed little to no sensitivity to variability in root biomass across grid cells and the ECM cost of N acquisition was always lower than the AM cost of N acquisition even in high N biomes. We have updated this parameter so that the updated CLM-FUN is equally sensitive to both availability and access, and can better capture latitudinal gradients in the benefit of ECM uptake or AM uptake as N becomes more limiting. This adjustment also ensures that while ECM plants invest more C belowground, they get a greater return on this investment relative to AM-associated plants when the ratio of N needed to support NPP to available soil N increases (e.g., enhanced N limitation under elevated $CO_2$) (Terrer *et al.* 2017). Specifically, we modified an AM-related uptake parameter and an ECM-related uptake parameter from $2.7 \times 10^{-4}$ (g C m$^{-2}$) to 6.2 (g C m$^{-2}$) and from $1.6 \times 10^{-3}$ (g C m$^{-2}$) to 34.1 (g C m$^{-2}$), respectively (see Table S1 for original and updated parameters). This parameter adjustment also resulted in small increase in the downregulation of NPP by FUN in CLM by 1.5 Pg C yr$^{-1}$ or ~3% for the last ten years of the model simulations from 1995-2004 (Figure S1). For this parameter adjustment, the spin-up, meteorological conditions, and time period are the same as outlined for CLM in Section 2.2 below.

To investigate the root symbiont associated C–climate feedback, we also used Community Atmosphere Model version 4 (CAM), an atmospheric general circulation model that includes CLM (or CLM-FUN) (Neale *et al*. 2010). CAM dynamically predicts

the impacts of external forcing factors such as anthropogenic $CO_2$ emissions on global and
regional climate (i.e., temperature and precipitation) by dynamically representing key
atmospheric process including cloud formation, aerosol impacts, radiative processes, and
mixing (Neale *et al*. 2010).
**2.2 Experimental Design**
In the first step of our model experiment, we leveraged the ability of FUN to
downregulate NPP in order to calculate the extent to which mycorrhizal fungi impact the
balance of C in the atmosphere vs. plant biomass. We estimated this by calculating the
difference in NPP between CLM runs with FUN turned on or off using the same
meteorological forcing data (Qian *et al.* 2006). The surface condition and plant functional
type (PFT) data are from the standard release of CLM4.0. The surface spin-up conditions,
in which the plant and soil C pools are at a quasi-equilibrium state, are provided with
CLM4.0 by the National Center for Atmospheric Research (NCAR). As such, both models
started from the same baseline values. We ran both CLM and CLM-FUN at the 0.9°×1.25°
and half-hourly spatio-temporal resolution for 25 years (1980–2004). The ambient $CO_2$
concentration was fixed to 338 ppm, the atmospheric $CO_2$ level in 1980. We calculated the
mean annual NPP difference between CLM and CLM-FUN in 1995–2004, and the value
was 8.1 Pg C $yr^{-1}$. This additionally respired C from CLM-FUN represents the C amount
that plants expend to take up N and we assumed that the all of this C went into the
atmospheric pool. We then converted this mass of extra C going into the atmosphere into
concentration units by dividing our mass (8.1Pg C) by the mass of C in 1ppm of $CO_2$ (2.135
Pg C). As such, we assume that integrating the C costs for N acquisition leads to an
additional 8.1 Pg C $yr^{-1}$of C released to the atmosphere at a 3.8 ppm of $CO_2$ annual rate.
Second, we ran two simulations of the land–atmosphere model, CAM4.0-CLM 4.0:
(1) A control simulation without mycorrhizal impacts on atmospheric $CO_2$ or surface
energy budgets (herein CAM), and (2) a simulation that included mycorrhizal impacts on
atmospheric $CO_2$ as well as surface energy budgets (herein CAM-FUN). Due to the
complexity and computational cost of running the fully coupled C-climate model, it was
necessary to prescribe the increase in $CO_2$ concentrations in CAM-FUN at a 3.8ppm
increase per year to reflect the transfer of C from NPP to the atmosphere. The CAM runs
did have dynamic representations of how the C cost of N acquisition impacted leaf area
index (LAI), evapotranspiration (ET), and resulting energy budgets. We acknowledge that
this assumption simplifies many of the interactions between the land, atmosphere and
ocean C pools.  However, given that our objective was to provide a first approximation of
how the C cost of N acquisition could impact climate, the prescribed $CO_2$ increase provides
a balance between meeting that objective and minimizing computational costs.  We used
the specified modern climatological sea surface temperatures and sea ice distributions and
ran the models at the 0.9°×1.25° and half-hourly spatio-temporal resolution for 25 years
from 1980-2004. In CAM, the ambient $CO_2$ concentration was 338 ppm. For all other
model inputs, we used the default input files that are automatically loaded during each
model run, such that both CAM and CAM-FUN start off with the same initial conditions.
In CAM-FUN, we assumed that atmospheric $CO_2$ started increasing from 338 ppm at the
3.8 ppm of $CO_2$ annual rate, and all the respired $CO_2$ is mixed into the atmosphere
homogenously. We also present the means of the CAM-based results for the last 10
simulation years from 1995-2004. We evaluated the climate impacts resulting from
including the mycorrhizal dynamics into CAM by calculating the surface air temperature
and precipitation differences between CAM and CAM-FUN in different regions.
In this study, we also estimated the radiative forcing variations causing the climate
impacts. We did this in order to identify which factor, ET vs. LAI vs. enhanced atmospheric
$CO_2$, led to our observed shifts in climate. It also allowed us to identify if the three different
forcing factors had a cooling or warming effect on the climate. We use the reflected solar
radiation difference between CAM and CAM-FUN to estimate the radiative forcing
variations from surface albedo change due to shifts in LAI. The evapotranspiration (ET)
difference between these two model runs was used to estimate the radiative forcing from
ET variation. The radiative forcing from $CO_2$ increase was calculated with an empirical
equation (Myhre *et al.* 1998):
$$\Delta F = \alpha \, ln(\frac{C}{C_0}) \tag{1}$$
where $\alpha$ is estimated as 5.35 (W m$^{-2}$), $C$ is $CO_2$ in parts per million by volume, and $C_0$ is the
reference concentration, which is 338 ppm, the atmospheric $CO_2$ level in 1980.
**3. Results**
Compared to the CAM runs where N was obtained at no cost, when we included
the C cost of symbiont-mediated N acquisition (i.e., CAM-FUN), C uptake by the terrestrial
biosphere was more strongly constrained by N availability. Consequently, N limitation
reduced global NPP by 2.4 g C m$^{-2}$ yr$^{-1}$, leading to alterations in atmospheric $CO_2$, global leaf
area index (LAI; Figures 1a and 1b), and surface energy budgets (Figure 2). Globally, NPP
and LAI were affected similarly, with the strongest relative effects occurring at the poles
and the strongest absolute effects occurring near the equator. In addition, we analyzed
temperature and precipitation shifts across three key biome classes that are delineated in
Figure S4. In boreal and alpine ecosystems, LAI was reduced by 34% (a decrease of 0.05
m$^2$ m$^{-2}$) while NPP was reduced by 42% (a decrease of 12 g C m$^{-2}$ yr$^{-1}$). In mid-latitude
temperate ecosystems, LAI was reduced by 17% (a decrease of 0.16 m$^2$ m$^{-2}$) while NPP was
reduced by 33% (a decrease of 30 g C m$^{-2}$ yr$^{-1}$). Tropical forest ecosystems had the largest
absolute reductions in LAI (0.24 m$^2$ m$^{-2}$; 10% decrease) and NPP (53 g C m$^{-2}$ yr$^{-1}$; 22%
decrease). Compared to NPP and LAI, ET had a more heterogeneous spatial pattern with a
global mean ET reduction 7.3 mm yr$^{-1}$, which represents a ~3% decrease across all of the
ecosystems (Figure 1c). While we present differences between model runs in LAI, ET and
NPP in Figure 1, global maps of the absolute values are presented in Figures S2 & S3.
Elevated $CO_2$ due to the reduction in NPP was the strongest driver of climate shifts.
The global NPP reduction (8.1 Pg C yr$^{-1}$) from the land model simulations resulted in an
increase in atmospheric $CO_2$ concentrations of 3.8 ppm yr$^{-1}$, and ~95 ppm over a 25-year
simulation. Accounting for the C cost of N acquisition in CAM's representation of N
limitation led to a net warming effect of 1.11 W m$^{-2}$ (Figure 2). By contrast, there was an
opposing effect of differences in LAI due to modifications of ET and surface albedo of the
vegetated land surface, leading to an overall net cooling effect of -0.52 W m$^{-2}$ (Figure 2).
The reduction in ET led to a cooling effect because it resulted in less water vapor in the
atmosphere which is a potent greenhouse gas. Integrated globally, these two opposing
effects led to a net warming effect of 0.59 W m$^{-2}$ (Figure 2), which resulted in a net increase
in surface air temperature by 0.1 °C and a net decrease in precipitation by 6 mm yr$^{-1}$,
globally.
While the averaged global impact of the C cost of microbial symbionts on climate
was minor (i.e., 0.1 °C surface air temperature increase and 6 mm yr⁻¹ precipitation decrease),
there were strong regional impacts in key biomes, particularly in forested regions with
ECM fungi (Figure 3). Moreover, the regional shifts in temperature were stronger those of
precipitation with shifts in precipitation being much more variable and patchier than those
of temperature (Figure 3). Given difficulties in predicting regional precipitation as well as
the high variability in our estimates, we present the data but acknowledge that these
regional estimates are uncertain. ECM-dominated areas in boreal and alpine biomes
became warmer (increases in surface air temperature by 1.0 °C) and wetter (increases in
precipitation by 9 mm yr⁻¹). Temperate forest ecosystems, which include plants that possess
all three nutrient acquisition strategies, were also impacted. The eastern part of North
America, Europe, and China had surface air temperature increases of 0.5 °C, and
precipitation shifted by 11, -37, and 2 mm yr⁻¹ in these three regions, respectively. Tropical
forests, which are dominated by AM fungi, were impacted less with temperature; Amazon
and Congo basin both had temperature increase by ~0.3 °C. However, precipitation
changes in tropical forests varied, with the Amazon and Congo basins drying by 4 mm yr⁻¹
and 49 mm yr⁻¹, respectively.
**4. Discussion and Conclusions**
Here, we demonstrate that integrating the C cost of N acquisition into the
formulation of N limitation in CAM reduced global NPP, LAI, and ET, with the greatest
percentage decreases in boreal and alpine ecosystems (Figure 1). These reductions led to
substantial impacts on climate, particularly in boreal and alpine ecosystems where
temperature increased by 1°C and precipitation increased by 9 mm yr⁻¹ over the last ten
years of the simulations (1995-2004) (Figure 3). It is important to note, that the regional
impacts of the C cost of N acquisition on temperature were much stronger than those on
precipitation.  These results suggest that by reducing C stored in woody biomass, the C
transferred to symbionts leads to more atmospheric $CO_2$ that would otherwise be locked up
in vegetation (Figure 2). This reduction in terrestrial productivity (Figure 1a) and decrease
of terrestrial C sink in CAM-FUN appears to alter the partitioning of energy fluxes at the
land surface into sensible heat flux as well, which accelerates land-surface warming and
intensified regional land–atmosphere feedback (Jung *et al*. 2010). Collectively, these
results suggest that the C cost of symbiont-mediated N acquisition is an important
component of the Earth's climate system that has the potential to alter future climate
trajectories.
The C expended by plants to support symbiont-mediated N uptake reduced the
amount of C available to support leaf growth and thus, reduced LAI. This global reduction
in LAI (Figure 1b) indirectly influenced climate through energy balance (i.e., albedo and
ET) feedbacks (Buermann *et al*. 2001). It has been suggested that changes in the
atmospheric heating pattern in the tropics as a result of the variations in latent heat flux
may modify the Hadley circulation, which then can change the generation of waves along
the polar front (Chase *et al*. 1996). As such, tropical LAI shifts (Figure 1b) can potentially
affect mid- and high-latitude climates and nearby ocean conditions through atmospheric
teleconnections (Feddema *et al*. 2005), a possible explanation for the greater climate
alterations we observed at high-latitudes.
We found greater spatial heterogeneity in ET shifts than NPP or LAI shifts when
we included the C cost of microbial symbionts in the model (Figure 1). Some of this spatial
variability may reflect the high sensitivity of ET to increases in atmospheric $CO_2$
concentrations (Shi *et al*. 2013). Moreover, this variability likely reflects the large
uncertainties and challenges associated with simulating regional scale ET in coupled
climate–atmosphere models (Boé & Terray, 2008; Pan *et al*. 2015). However, on the global
scale, the reduction of ET, which decreased the atmospheric concentration of water vapor,
a potent greenhouse gas, led to a -0.52 W m$^{-2}$ radiative forcing change (Figure 2). This
result is consistent with the Institut Pierre Simon Laplace climate model (IPSL-CM4)
(Davin *et al*. 2007), where ET was also reduced globally and had a net cooling effect on
global temperatures. Nevertheless, this cooling effect was outweighed by the warming
effect of increasing atmospheric $CO_2$ concentrations in CAM-FUN.
Our results suggest that models that do not account for plant-microbe symbiotic
interactions and the C cost of N acquisition may underestimate both N limitation to NPP
and rates of climate change. Nutrient limitation remains a key area of uncertainty for ESMs
with the CMIP5 comparison highlighting the limited representations of N limitation as a
primary reason for mismatch between the models and the observed C sink (Anav *et al*.
2013). Additionally, CAM-FUN identifies an important underestimation of nutrient
limitation and climate shifts in boreal and alpine ecosystems that has the potential to
enhance other climate feedbacks. Boreal forests, which dominate high-latitude regions, are
characterized by low rates of soil decomposition and low N availability (Read *et al*. 2004).
This leads to CAM-FUN predicting that boreal forests expend nearly 18% of NPP to gain
N through symbionts, a result that is supported by a recent empirical synthesis which found
that boreal forests have a 13-fold greater C cost of soil resource acquisition than tropical
forests (Gill & Finzi, 2016). However, to the extent that the greater C cost to ECM plants
(relative to AM plants) provides a greater return on investment of N under elevated $CO_2$
(Terrer *et al*. 2017), some of the predicted warming may be attenuated over time.
Nevertheless, predicted acceleration of warming in boreal forests is likely to be
consequential given feedbacks between surface warming with sea ice cover loss, sea
surface temperature increase, and permafrost thaw (Parmentier *et al*. 2013).
While CAM-FUN identifies an important interaction between the C cost of N
limitation and climate, there still remain key uncertainties in the model on the extent to
which other processes that govern the C cost of acquiring soil resources impacts C-climate
feedbacks. First, not all ecosystems are predominantly N limited (Wang *et al*. 2010). Nearly
30% of terrestrial ecosystems are limited by phosphorus (P) or water (Elser *et al*., 2007,
Fisher *et al*., 2010, Wieder *et al*., 2015).  These are two key limitations that are currently
absent from the model that may alter the climate trajectories shown here, particularly for
strongly P-limited ecosystems like tropical forests or water-limited ecosystems like
Mediterranean forests. However, FUN utilizes a modular structure based on optimal
allocation theory that could incorporate the C costs of P or water acquisition on NPP and
hence climate. As such, the optimal allocation parameterization in FUN could be modified
to include other resource costs and thus provides a framework for ESMs to assess how
multiple resource limitation impacts climate.
Second, the climate impacts we identify are sensitive to factors that alter N
availability. Across many ecosystems, increasing soil temperatures that enhance
decomposition (Melillo *et al*. 2011) or rising rates of N deposition in developing countries
(Liu *et al*. 2013) could increase N availability and lower the C cost of N acquisition.
Moreover, as currently formulated, the model omits important feedbacks between C
allocation to mycorrhizal symbionts and their ability to upregulate soil enzyme production,
prime soil organic matter decomposition and increase N availability (Brzostek *et al*. 2015,
Cheng *et al*. 2014, Finzi *et al*. 2015). A recent effort to couple FUN to a microbial soil
enzyme model at the ecosystem scale has shown that the ability of ECM fungi to prime
soil organic matter allowed them to mine N at the expense of soil C stocks to a greater
extent under elevated $CO_2$ than AM fungi (Sulman *et al*. 2017). This result is consistent
with recent meta-analyses that show that even though ECM plants invest more C
belowground than AM plants, they receive a greater N return on their investment under
elevated $CO_2$ (Terrer *et al*. 2017). As such, integrating C and N feedbacks between plant
and symbiotic microbes at the global scale represents a critical area for future model
development.
Finally, we acknowledge that the simplification of land-atmosphere interactions in
our model experiment may have precluded our ability to examine fully coupled feedbacks
that may have stimulated the land or ocean C sink.  This simplification was needed owing
to the complexity and computational resources needed to run the fully coupled model.  As
such, our estimates of the sensitivity of climate to the C cost of N acquisition likely
represents an upper bound.  This is due to two reasons. First, we assumed that all of the
carbon not sequestered as NPP was released into the atmosphere as $CO_2$.  In a fully coupled
model, it is likely that a portion of this $CO_2$ would have been sequestered by the ocean.
Second, the reduction of NPP due to the C cost of N acquisition also reduced heterotrophic
respiration by 3.3 Pg C yr$^{-1}$.  However, both empirical and modeling evidence suggests that
C expended belowground to gain N leads to greater soil organic matter decomposition and
respiration due to priming effects (Brzostek *et al*. 2015; Sulman *et al*. 2017). Lastly,
compared to other ESMs included in the Fifth Phase of the Coupled Model Intercomparison
Project (CMIP5), the land C pool in CESM/CLM4 is underestimated (Hoffman *et al*. 2014),
associated with a high-biased N downregulation and short turnover times for decomposing
C (Koven *et al*. 2014). This low-biased land C pool indicates an overestimation of the
atmospheric $CO_2$ burden over the 20th century (Hoffman *et al*. 2014).  Despite our
assumptions, experimental design, and bias impacting the model's ability to predict
absolute numbers, our modeling experiments allowed us to make the first test of the
sensitivity of the Earth's climate system to plant-microbial interactions.
To fully integrate the C cost of multiple soil resource acquisition into ESMs, there
are key empirical gaps that still need to be addressed including advancing observational
datasets of the distribution of nutrient acquisition strategies at the global scale and
expanding the spatial coverage and enhancing the temporal resolutions of both *in-situ* and
remote sensing data that can better parameterize the C cost of nutrient acquisition as well
as the N benefit of microbial symbionts (Fisher *et al*. 2016). This study shows that high-
latitude regions with low N available are more impacted by C cost of N acquisition.
However, remote sensing observations are limited in high-latitudes regions as a result of
the long snow-cover season and cloud contamination, Thus, *in-situ* and aircraft data can
potentially provide more accurate information in high-latitude regions. Given that the next
version of CESM will include the optimal allocation theory of FUN, addressing these
empirical and modeling gaps will aid in reducing uncertainty in the extent to which nutrient
limitation drives C–climate feedbacks.
**5. Data and Code Availability:** The data for all three figures as well as the model code
are available at: https://github.com/coffeesmj/Biogeosciences-Submission.git

**6. Acknowledgments.** Funding was provided by the US Department of Energy (Office of
Biological and Environmental Research, Terrestrial Ecosystem Science Program) and the
US National Science Foundation (Division of Environmental Biology, Ecosystem Studies
Program. The computations were performed at the Jet Propulsion Laboratory and at the
National Aeronautics and Space Administration (NASA) Ames Research Center. Junjie
Liu assisted with the computational resources. MS and JBF carried out the research at the
Jet Propulsion Laboratory, California Institute of Technology, under a contract with NASA,
and at the Joint Institute for Regional Earth System Science and Engineering, University
of California at Los Angeles. Government sponsorship acknowledged. Copyright 2018. All
rights reserved.
**7. Author contributions**
M.S. and E.R.B designed the research; M.S. conducted the model simulations and
performed the analyses; E.R.B and J.B.F contributed essential ideas of analyzing the
results; E.R.B and M.S wrote the manuscript with contributions from J.B.F. and R.P.P.

**8. Competing interests**
The authors declare no competing financial interests.
**9. Figures**

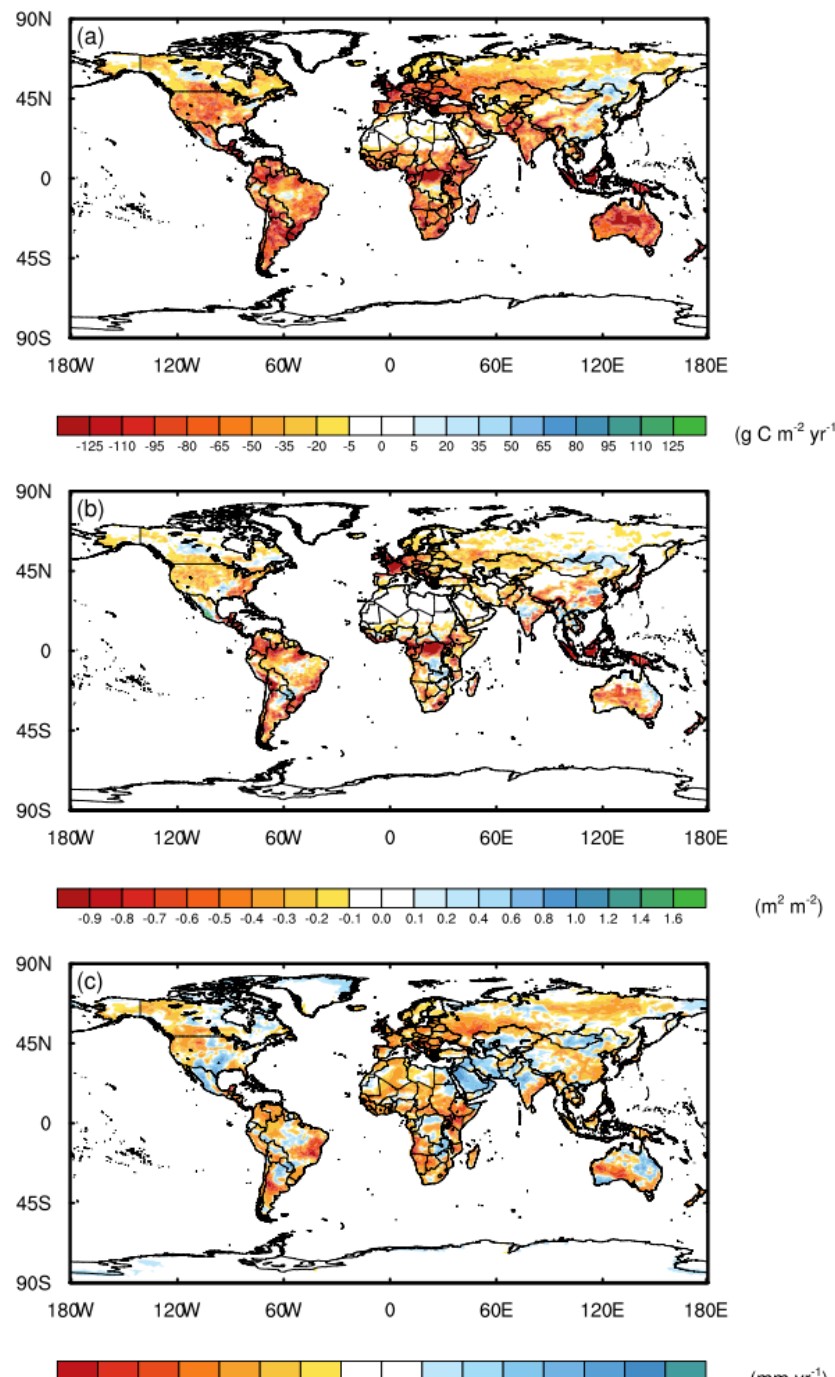

**Figure 1.** The C expended on symbiont-mediated N acquisition altered the spatial patterns
of (a) NPP, (b) LAI and (c) ET. These results were obtained from CAM runs with and
without the symbiont sub-module (CLM-FUN) and represent the mean of the last 10 years
of the simulation from 1995-2004.

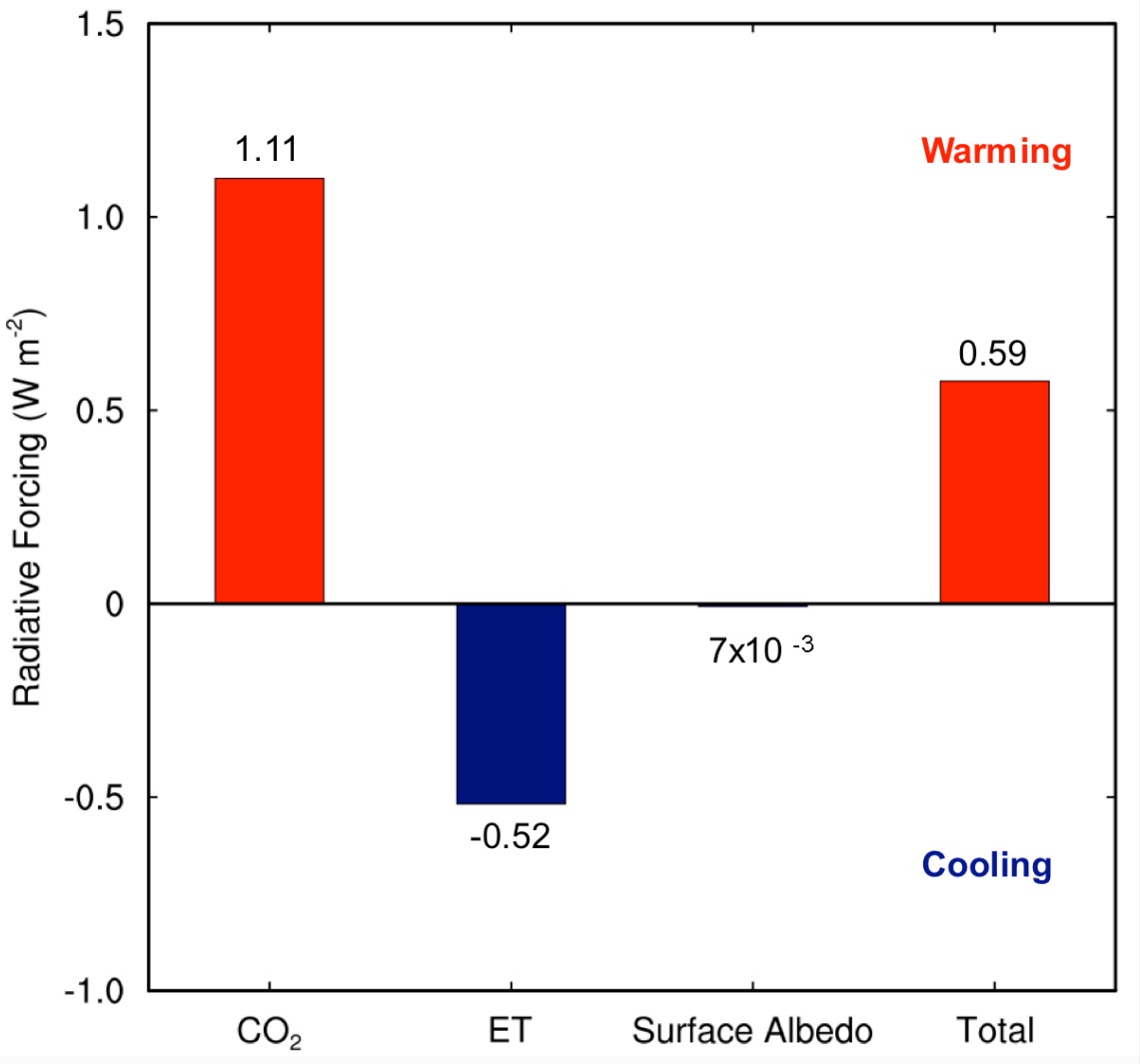

**Figure 2.** The impacts of the C cost of symbiont-mediated N acquisition led to a net
increase in global radiative forcing. The warming due to increasing atmospheric $CO_2$ was
offset partially by cooling due to reduced evapotranspiration (ET) and surface albedo.
These results were obtained from CAM runs with and without the symbiont sub-module
(CLM-FUN) and represent the mean of the last 10 years of the simulation from 1995-2004.

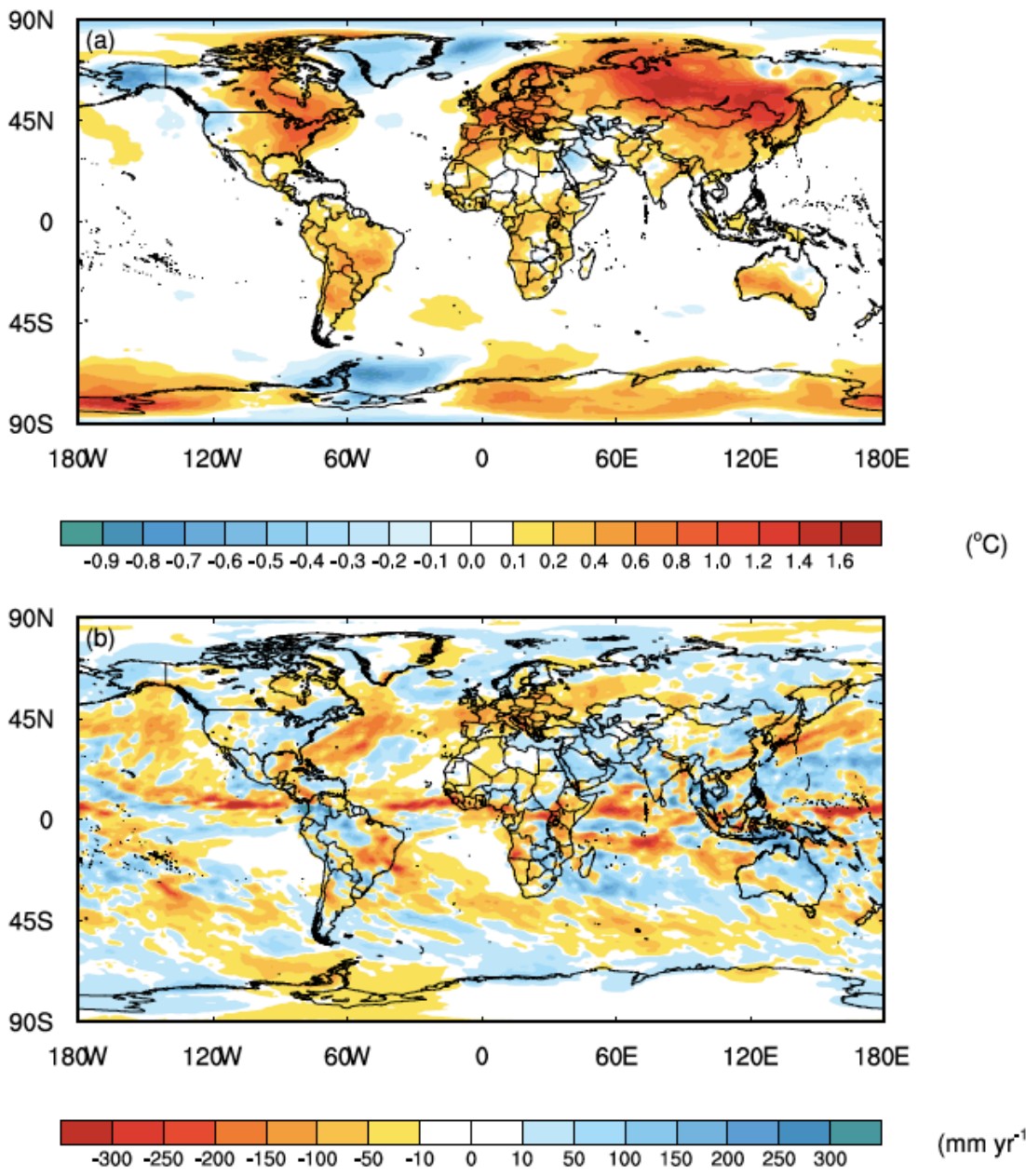

**Figure 3.** Feedbacks between symbiont-mediated N acquisition and C have a direct impact
on global climate. (a) Surface air temperatures increase across much of the land surface;
whereas (b) precipitation patterns are more variable. The values represent the mean
differences for each grid cell between CAM-FUN with ramping $CO_2$ and the baseline CAM
for the last 10 years of the simulations from 1995-2004.

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
