# Peer review of "Neglecting Plant-microbe Symbioses Leads to Underestimation of Modeled Climate Impacts"

_Biogeosciences, 2018_

## Referee Comment (RC1) · Anonymous Referee #1 · 30 Jul 2018

General Comments

Shi et al. address the effect of nitrogen (N) limitation on the land carbon (C) uptake and climate change by estimating C costs for N acquisition. They coupled the Fixation and Uptake of Nitrogen (FUN) sub-model, which directly calculates C costs for different N acquisition strategies, to the Community Land Model (CLM) and run CLM, and CLM-FUN respectively, to estimate the reduction of net primary production (NPP) under N limitation, firstly. Secondly they used CAM, an atmospheric circulation model that includes CLM (or CLM-FUN), to take impacts on climate change into account. They show that C costs for N acquisition lower NPP and the Leaf Area Index globally, what has implications for the global C budget as well as for evapotranspiration and surface albedo. This influences the global radiative forcing and water balance and leads to

changes in surface temperature and precipitation. Shi et al. summarize that Earth System Models that do not take C costs for N acquisition into account might over-estimate the land C sink and following under-predict climate change, but they also emphasize that P and water limitations play a role as well.

Overall, I think this is an interesting study and fits thematically the scope of BG quite well, but I have some major issues regarding the built and implementation of the study that make the evaluation impossible.

Specific Comments

I think, the 'Material and Methods' section is to short. Based on the given information it is impossible to reproduce the study, because it is unclear, what the authors have actually done and under which conditions the simulations run. The 'Results' part is very short, too, and does not show any base values, but only absolute and relative changes between model simulations. These short preparations of the following discussion make it impossible to fully understand the study and evaluate the paper.

Comments on FUN model

I miss more information about the modified parameters for N uptake by ECM / AM infected roots. Changes in orders of magnitudes require some more information. Not only, why the parameters were changed, but also how this effects the results compared to previous studies and/or a sensitivity analysis.

Comments on CLM vs CLM-FUN simulations

Regarding CLM-FUN vs CLM a proper set-up description is missing. If the reader wants to reproduce the study, he needs to know, if there is a spin-up done and how, how the models are initialized etc. From given information it is impossible to know for example, whether both models (CLM and CLM-FUN) start with the same biomass, or if they already differ in the beginning of the analyzed period, because of different spin-up results.

Secondly, the models calculate a global NPP difference of 8.2 PgC/yr. As consequence biomass in CLM-FUN should be lower as in CLM and thus also heterotrophic respiration should change. Do the authors consider that? If not, they might over-estimate the effect of N acquisition costs. Same for just looking at the land surface. As soon as the global land C sink is lowered, the ocean will take up more C, and not all additional C will remain in the atmosphere to force climate change. Over all it is completely unclear, how the authors derive the yearly $CO_2$ increase of 3.8 ppm from the calculated NPP difference. Moreover the increase of 3.8 ppm per year just because of taking C costs for N acquisition into account seems very high compared to the actually measured atmospheric $CO_2$ growth rate, which is around 2 ppm per year. Hence the derivation of the yearly atmospheric $CO_2$ increase should be described very detailed.

Comments on CAM vs CAM-FUN simulations

First of all, it is unclear, whether FUN is actually coupled to CAM (and the abbreviation CAM-FUN indicates that somehow), or if only the additional C release to the atmosphere (3.8 ppm/yr), which is calculated be CLM-FUN, is added to the atmosphere of CAM.

If FUN is coupled to CAM, the reader needs again a proper set-up description as required already for CLM-FUN. For a fully coupled CAM-FUN model, I don't understand the reason for the atmospheric $CO_2$ increase, because that should evolve internally by itself.

If only the atmospheric $CO_2$ concentration in CAM-FUN is increased compared to CAM, the analyzed effects might rather depend on the $CO_2$ forcing of CAM in general than on C costs for N acquisition. From the manuscript it is unclear, if any other changed values/fluxes from CLM-FUN are introduced to CAM-FUN, for example NPP to outbalance the additional C input to the atmosphere or is the total amount of C in CAM-FUN increasing. If NPP in CAM-FUN is constrained by CLM-FUN, how does that influence vegetation dynamics and development in CAM-FUN under increased atmospheric CO2?

Besides all that, the introduction of the optional slab mixed-layer ocean model is misleading, since it is not used. Is it?

Technical Corrections

L111: CAM is introduced as abbreviation of CAM version 4. Is there any other name for CAM than 'an atmospheric general circulation model that includes CLM'?

L122-L125: double reference to forcing data set

L124: spatio-temporal vs L139 spatiotemporal
* * *

---

## Referee Comment (RC2) · Anonymous Referee #2 · 23 Aug 2018

This is a study of the implications of the fact that most (or all) conventional modeling studies do not represent the expenditure of energy (C) by plants on the uptake of N. A previously-developed model of plant uptake (FUN) is used with the CLM land surface model to estimate the reduction in NPP as a result of N acquisition. This reduction in terrestrial uptake of C is then converted to a corresponding increase in atmospheric $CO_2$ which is fed into the CAM atmospheric model. Simulations of CAM with and without the FUN sub-model are used to quantify the impacts of N acquisition on global climate.

Although the manuscript is well-written in terms of the language used, I have serious concerns over the methodology and the information presented. As such I suggest that it requires major revision before it would be acceptable for publication.

[Figure]

At the very least the manuscript needs to do a better job at explaining what has been done (and possible limitations), but it is also possible that further simulations are required (particularly to clarify if the signal is robust).

**General comments**

One of my main concerns is that I am not sure I understand what the authors did - there is a need for more material in the methods section. A series of complicated modeling systems has been used but few details of the configurations and simulations are provided. I am not looking for 100% reproducibility - that is very difficult to achieve unless the author's github site includes all the configuration files, which I haven't checked - but the paper should provide more details than it does. For example, what were the initial conditions, was there a spin-up phase, what additional inputs are provided?

The discussion of the results is also very brief with only 35 lines in the Results section.

**Specific comments**

Abstract - I would like this to be more quantitative and also give some indication of the nature (and limitations) of the experimental design (e.g. ramped $CO_2$). At present it highlights the changes in "high-latitude" temperature and precipitation, but there are no other numbers.

L67 - It might be useful to add a line or two about the approach used in most climate models, e.g. N is "free" and NPP is simply "snipped" to match the N availability, to contrast with the approach used in FUN.

Section 2.1 - I don't expect full details of CLM (those can presumably be found in the literature that is cited) but a brief overview would be useful, particularly for people who have little or no idea what a land surface model is.

L102 "we updated the parameters" - It appears that the values of two parameters were changed by about 4 orders of magnitude and this is justified by a description of how the new model is better, but I would like to see more detail/evidence/justification. I haven't

read all the literature cited for FUN but I am left wondering why it was necessary to adjust the parameters by so much - or is it just that the results are not very sensitive to these values? In this area it might also help if the previous work with FUN was summarised - e.g. this is what has been done and found using FUN (coupled with other models?) previously. Can we see "before and after" patterns of, say, NPP, to show the improvements produced by changing the parameter values? If possible the names of the altered parameters should also be given (even if it is possibly obvious to anyone who reads the cited papers).

L111 CAM - I think this stands for Community Atmosphere Model, which should be explained. "optional slab mixed-layer ocean model" - I'm not so bothered that it is optional, but I do want to know if it is used here. L137 suggests prescribed SSTs were used and if that means no slab model then don't mention it. Is it relevant that CLM and CAM are part of CESM? Again, if not, don't mention it.

Experimental Design - CLM - how was the initial state of CLM prescribed? Was there a spin up? Was land use change included? Again I'm not looking for every detail so that I can definitely reproduce the results, but the reader should get a pretty good idea of what was done - which they don't at present.

Experimental design - CAM - I think that CAM-FUN means CAM with CLM and FUN...but I am not 100% sure. Another possibility is that it means "CAM with extra CO2 calculated from offline runs of CLM-FUN". Either way it needs to be clarified. Why is $CO_2$ ramped up, why not just start from a higher value? I guess the point is that N-acquistion gradually leads to enhanced atmospheric CO2...but on the other hand that is not something that started in 1980 and, ideally, one might have started both runs from a pre-industrial CO2. Why is the full 8.2 Pg C $yr^{-1}$ added to the atmosphere? In reality only a fraction ( 40%) of anthropogenic emissions of CO2 remain in the atmosphere, with ocean drawdown a large part of the story, so one might expect that something similar would apply here. I'm a bit confused by the whole approach to $CO_2$ used here, and this is another aspect. From the description it appears that CO2

is prescribed and not interactive in CAM(-FUN) (i.e. CLM-calculated fluxes of C do not change the atmospheric $CO_2$) but this should be clarified. Do both CAM and CAM-FUN start with the same amount of vegetation? Clarify what fluxes CLM exchanges with CAM, what is prescribed and what is interactive. All in all the design has to be better explained and justified.

**Results**

Are the changes in modeled climate (particularly temperature and precipitation) statistically significant? It is many years since I was involved in a paper that presented changes in modeled climate, but at that time it was considered essential to use an ensemble of runs (e.g. using different initial states) to quantify internal variability, and maps of changes would indicate the statistical significance of the change at each location. The widespread areas of increased temperature in Fig.3a are consistent with the "expected" change and are likely "meaningful", but the much more patchy changes in precipitation (Fig.3b) are less obviously signal rather than noise. If there can be no estimate of significance I think the discussion of changes in atmospheric hydrology have to be couched in much less certain language, with the limitations of the method flagged up. This becomes even more important at regional level.

Fig.2 and related discussion - I am not very familiar with how radiative forcing is used or calculated, but I am confused by the discussion! How is the radiative forcing from reduced evaporation calculated? Is this just the reduction in the latent heat flux (W m$^{-2}$)? The caption "warming...was offset..by..reduced evapotranspiration" is rather confusing - with reduced evaporation one might expect increased sensible heat flux (all else being equal) which would have a warming effect. L224 suggests that the ET change resulted in reduced water vapor and implies that that is where the radiative forcing comes from. I think we need better discussion of the energy balance and clarification of the radiative forcing/mechanisms. It might be quite correct but I am sure many readers of Biogeosciences are not familiar with the ideas of radiative forcing.

I can see that the study represents a "first look" at the implications of the C cost of N uptake on modeled climate - but it is unclear whether the methodology used allows for a meaningful estimate of the impact. Improved description and justification of the experimental design would clarify this, and at least improve the reader's confidence in the design, but at present I am left wondering what the experiment with a relatively rapid ramping up of atmospheric $CO_2$ (3.8 ppm per year) from an arbitrary start year (1980) actually tells us about the "real world". The authors conceded in L276 that there might be limitations to their method but do not properly enlarge on this. Convince me and I will be happy!

**Further details**

Title - I don't like this. "Plant-microbe symbioses reveal underestimation" suggests that the symbioses were somehow active or involved in the study. I would rephrase it as something like "Neglecting symbioses leads to underestimation of modeled impacts...".

L153 - if the units of dF are $W\,m^{-2}$, those of alpha should be the same (not $g\,m^{-2}$).

---

## Author Comment (AC1) · 12 Oct 2018

We are grateful to the reviewer for their thoughtful comments and we have addressed them below. We have also included a revised manuscript file which is uploaded here so that the direct changes that we made to the manuscript.

General Comments Shi et al. address the effect of nitrogen (N) limitation on the land carbon (C) uptake and climate change by estimating C costs for N acquisition. They coupled the Fixation and Uptake of Nitrogen (FUN) sub-model, which directly calculates C costs for different N acquisition strategies, to the Community Land Model (CLM) and run CLM, and CLM- FUN respectively, to estimate the reduction of net primary production (NPP) under N limitation, firstly. Secondly they used CAM, an atmospheric

circulation model that includes CLM (or CLM-FUN), to take impacts on climate change into account. They show that C costs for N acquisition lower NPP and the Leaf Area Index globally, what has implications for the global C budget as well as for evapotranspiration and surface albedo. This influences the global radiative forcing and water balance and leads to changes in surface temperature and precipitation. Shi et al. summarize that Earth System Models that do not take C costs for N acquisition into account might over- estimate the land C sink and following under-predict climate change, but they also emphasize that P and water limitations play a role as well. Overall, I think this is an interesting study and fits thematically the scope of BG quite well, but I have some major issues regarding the built and implementation of the study that make the evaluation impossible.

Specific Comments I think, the 'Material and Methods' section is to short. Based on the given information it is impossible to reproduce the study, because it is unclear, what the authors have actually done and under which conditions the simulations run. The 'Results' part is very short, too, and does not show any base values, but only absolute and relative changes between model simulations. These short preparations of the following discussion make it impossible to fully understand the study and evaluate the paper.

ERB: In the materials and methods section, we have added significant detail on the model simulations, the initial conditions, and the assumptions that we made. This includes both a better introduction into what each model does, more detailed information on the coupled CAM-CLM and CAM-CLM-FUN runs, and the initial conditions and forcing data for the models. In the results section, we have included two new global map figures in the supplemental information that show the absolute values of NPP, LAI, and ET in CAM-CLM and CAM-CLM-FUN.

Comments on FUN model I miss more information about the modified parameters for N uptake by ECM / AM infected roots. Changes in orders of magnitudes require some more information. Not only, why the parameters were changed, but also how this effects

the results compared to previous studies and/or a sensitivity analysis.

ERB: The FUN model predicts the C cost of N acquisition from the soil by ectomycor-rhizal, arbuscular mycorrhizal, and nonmycorrhizal roots based upon root biomass (a proxy for access) and soil nitrogen concentrations (a measure of availability of N for plants to take up). Previously, the parameter controlling the sensitivity of the C cost of N acquisition to root biomass was low. As such the C cost of N acquisition showed little to no sensitivity to variability in root biomass across grid cells and the ECM cost of N acquisition was always lower than the AM cost of N acquisition even in high N biomes. We have included a figure in the supplementary material that shows how modeled NPP changes with the new parameters as well as a table that shows the parameter changes. The parameter adjustment reduces global NPP by 1.5Pg or ~3%. Finally, we include text above in the material and methods in lines 130-149 that discusses this figure and the rationale behind the parameter adjustment.

Comments on CLM vs CLM-FUN simulations Regarding CLM-FUN vs CLM a proper set-up description is missing. If the reader wants to reproduce the study, he needs to know, if there is a spin-up done and how, how the models are initialized etc. From given information it is impossible to know for example, whether both models (CLM and CLM-FUN) start with the same biomass, or if they already differ in the beginning of the analyzed period, because of different spin-up results.

ERB: We have added text in lines 178-182 that states the model spinup and config-uration files are the default inputs that NCAR provides with the model. Both model configurations thus start from the same initial conditions and then diverge as FUN downregulates NPP in CLM based upon the C cost of acquisition.

Secondly, the models calculate a global NPP difference of 8.2 PgC/yr. As consequence biomass in CLM-FUN should be lower as in CLM and thus also heterotrophic respira-tion should change. Do the authors consider that? If not, they might over-estimate the effect of N acquisition costs. Same for just looking at the land surface. As soon as the

global land C sink is lowered, the ocean will take up more C, and not all additional C will remain in the atmosphere to force climate change. Over all it is completely unclear, how the authors derive the yearly $CO_2$ increase of 3.8 ppm from the calculated NPP difference. Moreover the increase of 3.8 ppm per year just because of taking C costs for N acquisition into account seems very high compared to the actually measured atmospheric $CO_2$ growth rate, which is around 2 ppm per year. Hence the derivation of the yearly atmospheric $CO_2$ increase should be described very detailed.

ERB: In CLM-FUN heterotrophic respiration was 3.3 Pg C yr-1 less than in CLM without FUN. This represents about 40% of the reduction we observed in NPP. While not including this reduction in heterotrophic respiration may impact the results we present, empirical and modeling evidence suggests that including the C cost of N acquisition likely enhances heterotrophic respiration. This would occur through C being sent belowground to rhizosphere microbes which enhances their ability to prime soil organic matter decomposition. This is why we highlighted the coupling of FUN with a microbial decomposition model in lines 368-370. We have also added text in the discussion to highlight this assumption as well as the lack of ocean uptake of $CO_2$ in the model in lines 383-391. We also acknowledge that our estimate of climate impacts represents an upper boundary condition that may be mediated by heterotrophic respiration and ocean processes. Also note that there was a rounding error in the NPP reduction and we have revised it down to 8.1 Pg C yr-1.

In addition, we assume that all of the C from the reduction of NPP is transferred to the atmosphere. We have included more detail on this calculation in lines 195-204. We have added text to the discussion to describe how these assumptions influence our results.

Comments on CAM vs CAM-FUN simulations First of all, it is unclear, whether FUN is actually coupled to CAM (and the abbreviation CAM-FUN indicates that somehow), or if only the additional C release to the atmo- sphere (3.8 ppm/yr), which is calculated be CLM-FUN, is added to the atmosphere of CAM. If FUN is coupled to CAM, the

reader needs again a proper set-up description as required already for CLM-FUN. For a fully coupled CAM-FUN model, I don't understand the reason for the atmospheric CO2 increase, because that should evolve internally by itself.

ERB: Due to complexity of running the fully coupled model of CAM with CLM in which the terrestrial biosphere impacts on C cycling dynamically interact with the atmosphere, we instead used an offline CLM-FUN run to calculate in experiment 1 the down regulation in NPP and assumed that this carbon that did not go into biomass instead went into the atmosphere. In experiment 2, we then run CAM with CLM or CLM-FUN. We then prescribe a CO2 increase in CAM-FUN and compare it to CAM with CLM only. Despite the lack of C cycling coupling, the resulting impacts of LAI or ET on energy budgets does influence radiative forcing. We have added text to clarify and justify this approach in lines 192-215 as well as text to state that CAM and CAM-FUN start off with the same initial conditions.

If only the atmospheric CO2 concentration in CAM-FUN is increased compared to CAM, the analyzed effects might rather depend on the CO2 forcing of CAM in general than on C costs for N acquisition. From the manuscript it is unclear, if any other changed values/fluxes from CLM-FUN are introduced to CAM-FUN, for example NPP to outbalance the additional C input to the atmosphere or is the total amount of C in CAM-FUN increasing. If NPP in CAM-FUN is constrained by CLM-FUN, how does that influence vegetation dynamics and development in CAM-FUN under increased atmospheric CO2?

ERB: Due to the computational cost of running the fully coupled model, we used the prescribed CO2 approach to assess the sensitivity of the climate system to the C cost of N acquisition. However, in the CAM-CLM and CAM-CLM-FUN runs we were able to assess the impacts of changing climate conditions on NPP, LAI, and ET.

Besides all that, the introduction of the optional slab mixed-layer ocean model is misleading, since it is not used. Is it?

[Figure]

ERB: We did not use the slab mixed-layer ocean model and have removed this from the text.

Technical Corrections L111: CAM is introduced as abbreviation of CAM version 4. Is there any other name for CAM than 'an atmospheric general circulation model that includes CLM'?

ERB: We have clearly defined CAM as the Community Atmosphere Model and have provided text to describe its function in lines 150-171.

L122-L125: double reference to forcing data set L124: spatio-temporal vs L139 spatiotemporal

ERB: We have corrected these errors.

Please also note the supplement to this comment:
https://www.biogeosciences-discuss.net/bg-2018-293/bg-2018-293-AC1-
supplement.pdf

---

## Author Comment (AC2) · 12 Oct 2018

ERB: We are grateful to the reviewer for their thoughtful comments and suggestions for improving the manuscript. In the supplement, we include a track change document that shows all of the edits and revisions that we made to the manuscript.

This is a study of the implications of the fact that most (or all) conventional modeling studies do not represent the expenditure of energy (C) by plants on the uptake of N. Apreviously-developed model of plant uptake (FUN) is used with the CLM land surface model to estimate the reduction in NPP as a result of N acquisition. This reduction in terrestrial uptake of C is then converted to a corresponding increase in atmospheric CO2 which is fed into the CAM atmospheric model. Simulations of CAM with and without the FUN sub-model are used to quantify the impacts of N acquisition on global climate.

Although the manuscript is well-written in terms of the language used, I have serious concerns over the methodology and the information presented. As such I suggest that it requires major revision before it would be acceptable for publication.

At the very least the manuscript needs to do a better job at explaining what has been done (and possible limitations), but it is also possible that further simulations are required (particularly to clarify if the signal is robust).

General comments

One of my main concerns is that I am not sure I understand what the authors did – there is a need for more material in the methods section. A series of complicated modeling systems has been used but few details of the configurations and simulations are provided. I am not looking for 100% reproducibility - that is very difficult to achieve unless the author's github site includes all the configuration files, which I haven't checked – but the paper should provide more details than it does. For example, what were the initial conditions, was there a spin-up phase, what additional inputs are provided?

ERB: We have added more detail to the materials and methods section to address these concerns. We now have text in lines 170-174 and 199-201 that states that the initial conditions, spinup configuration and other necessary conditions needed to run the simulations for both CLM and CAM. All of the configuration and spinup files are the default model inputs that are provided by the National Center for Atmospheric Research. We have also clearly defined the scope of our model experiments to alleviate confusion regarding the coupling of the two models.

The discussion of the results is also very brief with only 35 lines in the Results section.

ERB: In response to Reviewer 1, we have added in global maps of the absolute values of NPP, ET, and LAI for CAM-CLM with and without FUN to the Supplementary Material in lines 244-256. We have also highlighted the stronger impacts of the C cost of N acquisition for temperature than precipitation and greater uncertainty in precipitation estimates in lines 270-279.

Specific comments

Abstract - I would like this to be more quantitative and also give some indication of the nature (and limitations) of the experimental design (e.g. ramped CO2). At present it highlights the changes in "high-latitude" temperature and precipitation, but there are no other numbers.

ERB: We have added in more quantitative information into the Abstract. In addition, we have added a sentence that describes the experimental design.

L67 - It might be useful to add a line or two about the approach used in most climate models, e.g. N is "free" and NPP is simply "snipped" to match the N availability, to contrast with the approach used in FUN.

ERB: We have added text in lines 76-78 to state how typical climate models work per the reviewer's suggestion. In addition, we have added text to lines 80-82 to show how our previous work with FUN contrasts this common approach.

Section 2.1 - I don't expect full details of CLM (those can presumably be found in the literature that is cited) but a brief overview would be useful, particularly for people who have little or no idea what a land surface model is.

ERB: We have added text to provide a brief overview of the CLM and CAM models in lines 124-129 and lines 150-171, respectively.

L102 "we updated the parameters" - It appears that the values of two parameters were changed by about 4 orders of magnitude and this is justified by a description of how the new model is better, but I would like to see more detail/evidence/justification. I haven't read all the literature cited for FUN but I am left wondering why it was necessary to adjust the parameters by so much - or is it just that the results are not very sensitive to these values? In this area it might also help if the previous work with FUN was summarised - e.g. this is what has been done and found using FUN (coupled with other models?) previously. Can we see "before and after" patterns of, say, NPP, to show the improvements produced by changing the parameter values? If possible the names of the altered parameters should also be given (even if it is possibly obvious to anyone who reads the cited papers).

ERB: The FUN model predicts the C cost of N acquisition from the soil by ectomycorrhizal, arbuscular mycorrhizal, and nonmycorrhizal roots based upon root biomass (a proxy for access) and soil nitrogen concentrations (a measure of availability of N for plants to take up). Previously, the parameter controlling the sensitivity of the C cost of N acquisition to root biomass was low. As such the C cost of N acquisition showed little to no sensitivity to variability in root biomass across gridcells and the ECM cost of N acquisition was always lower than the AM cost of N acquisition even in high N biomes. We have included a figure in the supplementary material that shows how modeled NPP changes with the new parameters as well as a table that shows the parameter changes. The parameter adjustment reduces global NPP by 1.5Pg or ~3%. Finally, we include text above in the material and methods in lines 130-149 that discusses this figure and the rationale behind the parameter adjustment.

L111 CAM - I think this stands for Community Atmosphere Model, which should be explained. "optional slab mixed-layer ocean model" - I'm not so bothered that it is optional, but I do want to know if it is used here. L137 suggests prescribed SSTs were used and if that means no slab model then don't mention it. Is it relevant that CLM and CAM are part of CESM? Again, if not, don't mention it.

ERB: We have deleted this text from the materials and methods as we used prescribed sea surface temperatures as the reviewer noted and did not use the slab ocean model.

Experimental Design - CLM - how was the initial state of CLM prescribed? Was there a spin up? Was land use change included? Again I'm not looking for every detail so that I can definitely reproduce the results, but the reader should get a pretty good idea of what was done - which they don't at present.

ERB: We have added text in lines 178-182 that states the model spinup and configuration files are the default inputs that NCAR provides with the model. Both model configurations thus start from the same initial conditions and then diverge as FUN downregulates NPP in CLM based upon the C cost of acquisition.

Experimental design - CAM - I think that CAM-FUN means CAM with CLM and FUN...but I am not 100% sure. Another possibility is that it means "CAM with extra $CO_2$ calculated from offline runs of CLM-FUN". Either way it needs to be clarified. Why is $CO_2$ ramped up, why not just start from a higher value? I guess the point is that N-acquistion gradually leads to enhanced atmospheric $CO_2$...but on the other hand that is not something that started in 1980 and, ideally, one might have started both runs from a pre-industrial $CO_2$. Why is the full 8.2 Pg C yr-1 added to the atmosphere? In reality only a fraction ( 40%) of anthropogenic emissions of $CO_2$ remain in the atmosphere, with ocean drawdown a large part of the story, so one might expect that something similar would apply here. I'm a bit confused by the whole approach to $CO_2$ used here, and this is another aspect. From the description it appears that $CO_2$ is prescribed and not interactive in CAM(-FUN) (i.e. CLM-calculated fluxes of C do not change the atmospheric $CO_2$) but this should be clarified. Do both CAM and CAMFUN start with the same amount of vegetation? Clarify what fluxes CLM exchanges with CAM, what is prescribed and what is interactive. All in all the design has to be better explained and justified.

ERB: The reviewer is correct in how we configured the model runs. Due to complexity of running the fully coupled model of CAM with CLM in which the terrestrial biosphere impacts on C cycling dynamically interact with the atmosphere, we instead used an offline CLM-FUN run to calculate in experiment 1 the down regulation in NPP and assumed that this carbon that did not go into biomass instead went into the atmosphere. In experiment 2, we then run CAM with CLM or CLM-FUN. We then prescribe a $CO_2$

increase in CAM-FUN and compare it to CAM with CLM only. Despite the lack of C cycling coupling, the resulting impacts of LAI or ET on energy budgets does influence radiative forcing. We have added text to clarify and justify this approach in lines 195-204 as well as text in lines 207-209 to state that CAM and CAM-FUN start off with the same initial conditions.

Results

Are the changes in modeled climate (particularly temperature and precipitation) statistically significant? It is many years since I was involved in a paper that presented changes in modeled climate, but at that time it was considered essential to use an ensemble of runs (e.g. using different initial states) to quantify internal variability, and maps of changes would indicate the statistical significance of the change at each location. The widespread areas of increased temperature in Fig.3a are consistent with the "expected" change and are likely "meaningful", but the much more patchy changes in precipitation (Fig.3b) are less obviously signal rather than noise. If there can be no estimate of significance I think the discussion of changes in atmospheric hydrology have to be couched in much less certain language, with the limitations of the method flagged up. This becomes even more important at regional level.

ERB: Given that we did not do an ensemble of runs, we are not able to evaluate significance. As such, we have added text in the results in lines 273-282 to couch the precipitation results and to acknowledge the low signal to noise ratio in the precipitation results.

Fig.2 and related discussion - I am not very familiar with how radiative forcing is used or calculated, but I am confused by the discussion! How is the radiative forcing from reduced evaporation calculated? Is this just the reduction in the latent heat flux (W m-2)? The caption "warming...was offset..by..reduced evapotranspiration" is rather confusing - with reduced evaporation one might expect increased sensible heat flux (all else being equal) which would have a warming effect. L224 suggests that the ET change resulted in reduced water vapor and implies that that is where the radiative forcing comes from. I think we need better discussion of the energy balance and clarification of the radiative forcing/mechanisms. It might be quite correct but I am sure many readers of Biogeosciences are not familiar with the ideas of radiative forcing.

ERB: We have added text in the methods to explain why we were doing this analysis which lets us see which of these factors had the biggest impact on climate in lines 231-233 and also in the results to state that ET had a cooling effect due to reductions in water vapor in lines 267-268.

I can see that the study represents a "first look" at the implications of the C cost of N uptake on modeled climate - but it is unclear whether the methodology used allows for a meaningful estimate of the impact. Improved description and justification of the experimental design would clarify this, and at least improve the reader's confidence in the design, but at present I am left wondering what the experiment with a relatively rapid ramping up of atmospheric CO2 (3.8 ppm per year) from an arbitrary start year (1980) actually tells us about the "real world". The authors conceded in L276 that there might be limitations to their method but do not properly enlarge on this. Convince me and I will be happy!

ERB: We have increased the text describing the limitations as well as benefits of our approach in the Discussion in lines 383-391. In addition, we have made substantial changes to the methods to help clarify and justify our approach as highlighted in responses above.

Further details

Title - I don't like this. "Plant-microbe symbioses reveal underestimation" suggests that the symbioses were somehow active or involved in the study. I would rephrase it as something like "Neglecting symbioses leads to underestimation of modeled impacts...".

ERB: We have changed the title to: "Neglecting plan-microbe sysmbioses leads to underestimation of modeled climate impacts."

L153 - if the units of dF are W mô$\breve{A}\breve{A}$2, those of alpha should be the same (not g mô$\breve{A}\breve{A}$2).

ERB: We have corrected this mistake.

Please also note the supplement to this comment:
https://www.biogeosciences-discuss.net/bg-2018-293/bg-2018-293-AC2-supplement.pdf

**Supplement:**

[revised manuscript text omitted]

In this study, we also estimated the radiative forcing variations causing the climate impacts. We did this in order to identify which factor, ET vs. LAI vs. enhanced atmospheric $CO_2$, led to our observed shifts in climate. It also allowed us to identify if three different forcing factors had a cooling or warming effect on the climate. We used

Eddie Brzostek 10/11/2018 12:26 PM

Eddie Brzostek 10/11/2018 12:37 PM

Eddie Brzostek 10/11/2018 9:49 AM

Eddie Brzostek 10/11/2018 9:43 AM

Eddie Brzostek 10/11/2018 9:43 AM

Eddie Brzostek 10/11/2018 9:43 AM

Eddie Brzostek 10/11/2018 9:51 AM

Eddie Brzostek 10/11/2018 9:51 AM

Eddie Brzostek 10/11/2018 9:51 AM

Eddie Brzostek 10/12/2018 7:52 AM

Eddie Brzostek 10/12/2018 7:52 AM

Eddie Brzostek 10/11/2018 10:14 AM

Eddie Brzostek 10/11/2018 10:14 AM

Eddie Brzostek 10/11/2018 10:15 AM

Eddie Brzostek 10/11/2018 10:26 AM

Eddie Brzostek 10/11/2018 2:47 PM

the reflected solar radiation difference between CAM and CAM-FUN to estimate the radiative forcing variations from surface albedo change due to shifts in LAI. ET consists of canopy evaporation, canopy transpiration, and ground evaporation, and the radiative forcing of these three components is calculated by CLM. Thus, we summed up the radiative forcing of these three components, and calculated their difference between the two model runs to estimate the radiative forcing variation as a results of ET changes. The evapotranspiration (ET) difference between these two model runs was used to estimate the radiative forcing from ET variation. The radiative forcing from $CO_2$ increase was calculated with an empirical equation as (Myhre *et al*. 1998).

$$\Delta F = \alpha \, ln(\frac{C}{C_0}) \tag{1}$$

where $\alpha$ is estimated as 5.35 (W m$^{-2}$), $C$ is $CO_2$ in parts per million by volume, and $C_0$ is the reference concentration, which is 338 ppm, the atmospheric $CO_2$ level in 1980.

**3. Results**

Compared to the CAM runs where N was obtained at no cost, when we included the C cost of symbiont-mediated N acquisition (i.e., CAM-FUN), C uptake by the terrestrial biosphere was more strongly constrained by N availability. Consequently, N limitation reduced global NPP by 2.4 g C m$^{-2}$yr$^{-1}$, leading to alterations in atmospheric $CO_2$, global leaf area index (LAI; Figures 1a and 1b), and surface energy budgets (Figure 2). Globally, NPP and LAI were affected similarly, with the strongest relative effects occurring at the poles and the strongest absolute effects occurring near the equator. In high-latitude ecosystems, LAI was reduced by 34% (a decrease of 0.05 m$^2$ m$^{-2}$) while NPP was reduced by 42% (a decrease of 12 g C m$^{-2}$yr$^{-1}$). In mid-latitude temperate ecosystems, LAI was reduced by 17% (a decrease of 0.16 m$^2$ m$^{-2}$) while NPP was reduced by 33% (a decrease of 30 g C m$^{-2}$yr$^{-1}$). Tropical low latitude ecosystems had the largest absolute reductions in LAI (0.24 m$^2$ m$^{-2}$; 10% decrease) and NPP (53 g C m$^{-2}$yr$^{-1}$; 22% decrease). Compared to NPP and LAI, ET had a more heterogeneous spatial pattern with a global mean ET reduction 7.3 mm yr$^{-1}$, which represents a ~3% decrease across high, mid, and low latitude ecosystems (Figure 1c). While we present differences between model runs in LAI, ET and NPP in Figure 1, global maps of the absolute values are presented in Figures S2 & S3.

Elevated $CO_2$ due to the reduction in NPP was the strongest driver of climate shifts. The global NPP reduction (8.1 Pg C yr$^{-1}$) from the land model simulations resulted in an increase in atmospheric $CO_2$ concentrations of 3.8 ppm yr$^{-1}$, and ~95 ppm over a 25-year simulation. Accounting for the C cost of N acquisition in CAM's representation of N limitation led to a net warming effect of 1.11 W m$^{-2}$ (Figure 2). By contrast, there was an opposing effect of differences in LAI due to modifications of ET and surface albedo of the vegetated land surface, leading to an overall net cooling effect of -0.52 W m$^{-2}$ (Figure 2). The reduction in ET led to a cooling effect because it resulted in less water vapor in the atmosphere which is a potent greenhouse gas. Integrated globally, these two opposing effects led to a net warming effect of 0.59 W m$^{-2}$ (Figure 2), which resulted in a net increase in surface air temperature by 0.1 °C and a net decrease in precipitation by 6 mm yr$^{-1}$, globally.

While the averaged global impact of the C cost of microbial symbionts on climate was minor (i.e., 0.1 °C surface air temperature increase and 6 mm yr$^{-1}$ precipitation decrease), there were strong regional impacts in key biomes, particularly in forested regions with ECM fungi (Figure 3). Moreover, the regional shifts in temperature were stronger those of precipitation with shifts in precipitation being much more variable and patchier than those of temperature (Figure 3). Given difficulties in predicting regional precipitation as well as the high variability in our estimates, we present the data but acknowledge that these regional estimates are uncertain. The ECM-dominated boreal

Eddie Brzostek 10/11/2018 11:24 AM

Eddie Brzostek 10/11/2018 11:44 AM

Eddie Brzostek 10/11/2018 2:51 PM

Eddie Brzostek 10/11/2018 2:46 PM

Eddie Brzostek 10/11/2018 2:46 PM

[revised manuscript text omitted]

Unknown

Figure S2.  Absolute values of (a) NPP, (b) LAI, and (c) ET in CAM without CLM-FUN.

[Figure]

Figure S3. Absolute values of (a) NPP, (b) LAI, and (c) ET in CAM with CLM-FUN.

**Table S1.** The adjusted parameters in CLM-FUN

| Cost Parameter | Original | Updated |
| --- | --- | --- |
| $AK_C$ | $2.7 \times 10^{-4}$ | 6.2 |
| $AK_N$ | $5.5 \times 10^{-5}$ | $5.5 \times 10^{-5}$ |
| $EK_C$ | $1.6 \times 10^{-3}$ | 34.1 |
| $EK_N$ | $2.7 \times 10^{-4}$ | $2.7 \times 10^{-4}$ |
| $K_C$ | $5.5 \times 10^{-5}$ | $5.5 \times 10^{-5}$ |
| $K_N$ | $3.3 \times 10^{-3}$ | $3.3 \times 10^{-3}$ |
| $K_R$ | $8.0/4.4 \times 10^{-4}$ [b] | $8.0/4.4 \times 10^{-4}$ [b] |

[a] The parameter values used in Shi *et al*. [2016].

[b] 8.0 was used for deciduous plant functional types (PFTs) and $4.4 \times 10^{-4}$ was used for evergreen PFTs.

---

## Editor Decision (ED1)

**Second review of Shi et al., 2018, "Plant-microbe symbioses reveal underestimation of modeled climate impacts", Biogeosciences.**

In general I find that the revised manuscript is a substantial improvement on the original submission, and I consider that it is suitable for publication after minor corrections. In particular the "Material and Methods" section has had material added to better describe the series of model simulations. The revised title is also clearer.

Although the authors have introduced various caveats about the limited nature of their climate model runs (2 runs starting from the same initial conditions are compared over a 10 year period), some of the discussion is still rather limited in this respect (e.g. no explicit mention of internal variability of the model, nor of possible trends related to a common initial state). This might be a deliberate attempt to suit the intended audience, or might possibly reflect the authors' backgrounds and interests. Given the intended audience and the stated aim being to provide a first estimate of the implications of neglecting the C costs, I consider that the level of detail provided is probably sufficient. Although we need to be wary of the details of the 10yr (climate) changes described (wary of the precise quantification) I have no reason to doubt that the effects are real (possibly with different magnitude).

**Minor comments**

L89-91 "we imposed a simplification" – this sounds like you actively modified CAM but I think what is meant is that you did not use interactive ocean or sea ice models (rather you prescribed SSTs and ice amounts). Rephrase along the lines of: "we used CAM with prescribed sea surface temperatures and sea ice, and introduced symbiotic processes…".

L146 – Ideally we might also get more details of the experimental setup for these early simulations of CLM-FUN, though I suspect they followed the configuration used later in section 2.2. As a minimum I suggest adding something to indicate that the downregulation of NPP referred to contemporary or recent conditions, or 1995-2004, or whatever. Or if both sets of CLM simulations shared common details, move them to here .

L209 and following – Again, I assume that the CAM runs were for 1980-2004, as was used for CLM – but this should be stated. And so all later averages from CAM refer to the years 1995-2004?

L262 – What are the bounds (latitudes?) of the various areas used to calculate averages, e.g. high-latitude, mid-latitude, tropical low latitude? Add these after each, e.g. high-latitude ecosystems (60-85N). Otherwise we have these rather vague descriptions of areas next to precise statistics of changes.

L305 and others – I don't find these statistics of regional changes in precipitation convincing, or useful, given that they are based on only two 10-yr simulations. Personally I would consider removing much of L300-305. However, as the aim of this paper is to provide first estimates, not a detailed account of atmospheric changes, it is probably acceptable to leave the statistics in the manuscript.

L312 – I think this is the first mention of using 10yr statistics from CAM (see earlier request for clarity on this). Also rephrase along the lines of "temperature increased by 1degC over 10 years and precipitation increased by 9 mm yr-1".

L341 – -5.2 W m-2   should be -0.52 W m-2.

Figures – captions should say that they refer to 10 year averages (or whatever).

Table S1 – I can't see where footnote a is referenced. I suspect it applies to (at least) the middle column.

---

## Author Response (AR2)

We appreciate the reviewer's thoughtful and careful read of the revision of our manuscript. Below we have addressed the comments and additional issues that the reviewer has raised. Our responses will be indicated by **ERB et al.** followed by our response in *italics*. All line numbers referenced below refer to the manuscript version that includes the track changed markup.

**Second review of Shi et al., 2018, "Plant-microbe symbioses reveal underestimation of modeled climate impacts", Biogeosciences.**

In general I find that the revised manuscript is a substantial improvement on the original submission, and I consider that it is suitable for publication after minor corrections. In particular the "Material and Methods" section has had material added to better describe the series of model simulations. The revised title is also clearer.

Although the authors have introduced various caveats about the limited nature of their climate model runs (2 runs starting from the same initial conditions are compared over a 10 year period), some of the discussion is still rather limited in this respect (e.g. no explicit mention of internal variability of the model, nor of possible trends related to a common initial state). This might be a deliberate attempt to suit the intended audience, or might possibly reflect the authors' backgrounds and interests. Given the intended audience and the stated aim being to provide a first estimate of the implications of neglecting the C costs, I consider that the level of detail provided is probably sufficient. Although we need to be wary of the details of the 10yr (climate) changes described (wary of the precise quantification) I have no reason to doubt that the effects are real (possibly with different magnitude).

**ERB et al.:** *As highlighted by the reviewer, we have focused this manuscript primarily on meeting our objective to provide a first estimate on the impacts of neglecting the C costs of N acquisition by symbionts on climate. In the previous round of revisions, we have added text that addresses the uncertainties and caveats in our model estimates that we feel address the reviewer's concerns about the absolute magnitude of the changes in climate we present.*

**Minor comments**

L89-91 "we imposed a simplification" – this sounds like you actively modified CAM but I think what is meant is that you did not use interactive ocean or sea ice models (rather you prescribed SSTs and ice amounts). Rephrase along the lines of: "we used CAM with prescribed sea surface temperatures and sea ice, and introduced symbiotic processes…".

**ERB et al.:** *We have rephrased this sentence in the introduction in lines 88-91 following the reviewer's suggestion.*

L146 – Ideally we might also get more details of the experimental setup for these early simulations of CLM-FUN, though I suspect they followed the configuration used later in section 2.2. As a minimum I suggest adding something to indicate that the downregulation of NPP referred to contemporary or recent conditions, or 1995-2004, or whatever. Or if both sets of CLM simulations shared common details, move them to here.

**ERB et al.:** *We have added text to clarify the time period as well as the initial model state and spin-up in lines 138-142: "This parameter adjustment also resulted in small increase in the downregulation of NPP by FUN in CLM by 1.5 Pg C yr-1 or ~3% for the last ten years of the model simulations from 1995-2004 (Figure S1). For this parameter adjustment, the spin-up, meteorological conditions, and time period are the same as outlined for CLM in Section 2.2*

*below." We have chosen to leave the detailed model description in Section 2.2 as those model simulations represent the model simulations that were performed to meet the objectives of this study.*

L209 and following – Again, I assume that the CAM runs were for 1980-2004, as was used for CLM – but this should be stated. And so all later averages from CAM refer to the years 1995-2004?

**ERB et al.:** *To address this comment, we have added text in lines 188-190 that explicitly states the 10-year period from 1995-2004 that was analyzed for differences in climate. In addition, we have added text to the figure captions to clarify the time period over which we analyzed differences.*

L262 – What are the bounds (latitudes?) of the various areas used to calculate averages, e.g. high-latitude, mid-latitude, tropical low latitude? Add these after each, e.g. high-latitude ecosystems (60-85N). Otherwise we have these rather vague descriptions of areas next to precise statistics of changes.

**ERB et al.:** *We have added a new figure to the SI material that provides maps of the three key biome classes over which we analyzed the model results for regional changes in climate. In addition, we have added text in the results in lines 224-226 that point the reader to this figure.*

L305 and others – I don't find these statistics of regional changes in precipitation convincing, or useful, given that they are based on only two 10-yr simulations. Personally I would consider removing much of L300-305. However, as the aim of this paper is to provide first estimates, not a detailed account of atmospheric changes, it is probably acceptable to leave the statistics in the manuscript.

**ERB et al.:** *We have chosen to leave this text in the manuscript as we added text in the previous round of revisions that clearly states the uncertainty in these estimates in lines 243-245: "Given difficulties in predicting regional precipitation as well as the high variability in our estimates, we present the data but acknowledge that these regional estimates are uncertain".*

L312 – I think this is the first mention of using 10yr statistics from CAM (see earlier request for clarity on this). Also rephrase along the lines of "temperature increased by 1degC over 10 years and precipitation increased by 9 mm yr-1".

**ERB et al.:** *We have revised the text in lines 259-262 to make it explicit that this is referring to the differences in the last ten years of the model simulations.*

L341 – -5.2 W m-2 should be -0.52 W m-2.

**ERB et al.:** *We have corrected this error.*

Figures – captions should say that they refer to 10 year averages (or whatever).

**ERB et al.:** *We have added this information to all the figure captions in the main manuscript and SI material where this is applicable.*

Table S1 – I can't see where footnote a is referenced. I suspect it applies to (at least) the middle column.

**ERB et al.:** *We have corrected this error so that the footnote a refers to the middle column in the table.*

[revised manuscript text omitted]

11. Supplementary Materials

[Figure]

**Figure S1.** Impacts of parameter adjustments in FUN on predicted global NPP in CLM.
Values represent the mean difference between the original parameterization and the new
parameterization for the last ten years of the simulation from 1995-2004. On a global scale,
the new parameterization reduced NPP from 50.8 Pg C yr⁻¹ to 49.3 Pg C yr⁻¹.

[Figure]

Figure S2.  Absolute values of (a) NPP, (b) LAI, and (c) ET in CAM without CLM-FUN.
These values represent the mean of the last ten years of the simulation from 1995-2004.

[Figure]

Figure S3.  Absolute values of (a) NPP, (b) LAI, and (c) ET in CAM with CLM-FUN.
These values represent the mean of the last ten years of the simulation from 1995-2004.

[Figure]

Figure S4. Global maps of three key biome classes that were analyzed for regional shifts
in temperature and precipitation: (a) tropical forests, (b) temperate forests, and (c) boreal
and alpine forests. Areas were delineated by grid cells that contained greater than 25% of
each plant functional type.

**Table S1.** The adjusted parameters in CLM-FUN

| Cost Parameter | Original[a] | Updated |
|---|---|---|
| $AK_C$ | $2.7\times10^{-4}$ | 6.2 |
| $AK_N$ | $5.5\times10^{-5}$ | $5.5\times10^{-5}$ |
| $EK_C$ | $1.6\times10^{-3}$ | 34.1 |
| $EK_N$ | $2.7\times10^{-4}$ | $2.7\times10^{-4}$ |
| $K_C$ | $5.5\times10^{-5}$ | $5.5\times10^{-5}$ |
| $K_N$ | $3.3\times10^{-3}$ | $3.3\times10^{-3}$ |
| $K_R$ | $8.0/4.4\times10^{-4}$ [b] | $8.0/4.4\times10^{-4}$ [b] |

[a] The parameter values used in Shi *et al*. [2016].

[b] 8.0 was used for deciduous plant functional types (PFTs) and $4.4\times10^{-4}$ was used for evergreen PFTs.